# Crowdsourced analysis of fungal growth and branching on microfluidic platforms

**Alex Hopke**[1,2,3], **Alex Mela**[4¤a], **Felix Ellett**[1,2], **Derreck Carter-House**[5], **Jesús F. Peña**[5], **Jason E. Stajich**[5], **Sophie Altamirano**[6], **Brian Lovett**[7], **Martin Egan**[8], **Shiv Kale**[9], **Ilkka Kronholm**[10], **Paul Guerette**[11], **Edyta Szewczyk**[11], **Kevin McCluskey**[11], **David Breslauer**[11], **Hiral Shah**[12¤b], **Bryan R. Coad**[13], **Michelle Momany**[4]*, **Daniel Irimia**[1,2,3]*

**1** Center for Engineering in Medicine and Surgery, Massachusetts General Hospital, Boston, Massachusetts, United States of America, **2** Harvard Medical School, Boston, Massachusetts, United States of America, **3** Shriners Hospital for Children, Boston, Massachusetts, United States of America, **4** Fungal Biology Group and Plant Biology Department, University of Georgia, Athens, Georgia, United States of America, **5** Department of Microbiology and Plant Pathology, University of California, Riverside, California, United States of America, **6** Department of Microbiology and Immunology, University of Minnesota, Minneapolis, Minnesota, United States of America, **7** Division of Plant and Soil Sciences, West Virginia University, Morgantown, West Virginia, United States of America, **8** Department of Entomology and Plant Pathology, University of Arkansas, Fayetteville, Arkansas, United States of America, **9** Nutritional Immunology and Molecular Medicine Institute, Blacksburg, Virginia, United States of America, **10** Department of Biological and Environmental Science, University of Jyväskylä, Jyväskylä, Finland, **11** Bolt Threads Inc., Emeryville, California, United States of America, **12** Bharat Chattoo Genome Research Centre, Department of Microbiology and Biotechnology Centre, The Maharaja Sayajirao University of Baroda, Vadodara, India, **13** School of Agriculture, Food & Wine, University of Adelaide, Adelaide, South Australia, Australia

¤a Current address: University of California, Riverside, California, United States of America
¤b Current address: Cell Biology and Biophysics, European Molecular Biology Laboratory, Heidelberg, Germany
* dirimia@mgh.harvard.edu (DI); mmomany@uga.edu (MM)

**Data Availability Statement:** Most data are within the manuscript and its Supporting information files. The source data for Figs 2b, 2c, 5a–5d, 6a–6d, 7b and 7c are available in the "source data file".

## Abstract

Fungal hyphal growth and branching are essential traits that allow fungi to spread and proliferate in many environments. This sustained growth is essential for a myriad of applications in health, agriculture, and industry. However, comparisons between different fungi are difficult in the absence of standardized metrics. Here, we used a microfluidic device featuring four different maze patterns to compare the growth velocity and branching frequency of fourteen filamentous fungi. These measurements result from the collective work of several labs in the form of a competition named the "Fungus Olympics." The competing fungi included five ascomycete species (ten strains total), two basidiomycete species, and two zygomycete species. We found that growth velocity within a straight channel varied from 1 to 4 μm/min. We also found that the time to complete mazes when fungal hyphae branched or turned at various angles did not correlate with linear growth velocity. We discovered that fungi in our study used one of two distinct strategies to traverse mazes: high-frequency branching in which all possible paths were explored, and low-frequency branching in which only one or two paths were explored. While the high-frequency branching helped fungi escape mazes with sharp turns faster, the low-frequency turning had

The accession numbers for the invasive Aspergillus fumigatus and Rhizopus microsporus clinical isolates are withheld because they contain potentially identifying and sensitive patient information, according to the Central Adelaide Local Health Network Human Research Ethics Committee (CALHN HREC) provisions. Data requests may be sent to Dr. Sarah Kidd, National Mycology Reference Centre, SA Pathology, Frome Road, Adelaide, SA, 5000, Australia, sarah.kidd@sa.gov.au.

**Funding:** Funding to the Irimia lab included support from NIH GM092804 and EB002503. Funding to the Momany lab included support from the Burroughs Wellcome Fund CRT1017499. Funding to the Stajich lab included support from NSF DEB-1441715. Stajich is a CIFAR fellow in the program Fungal Kingdom: Threats and Opportunities. We would like to acknowledge the contribution of CytoSmart (www.cytosmart.com), who provided the microscopes and helped to defray the cost of shipping microscopes to participants.

**Competing interests:** The authors have read the journal's policy and have the following competing interests: Paul Guerette, Edyta Szewczyk, Kevin McCluskey, and David Breslauer are paid employees of Bolt Threads, Inc. There are no patents, products in development or marketed products associated with this research to declare. This does not alter our adherence to PLOS ONE policies on sharing data and materials.

a significant advantage in mazes with shallower turns. Future work will more systematically examine these trends.

## Introduction

Filamentous fungi are essential for industry, agriculture, and biomedical research [1]. They contribute to soil fertility, have a great capacity to make and secrete products, and hold enormous economic potential [2]. They are essential tools for studying fundamental biological questions, including the cell cycle and other cellular processes [3, 4]. Filamentous fungi are also important pathogens of plants, causing significant crop losses [5], and of animals, causing life-threatening infections. Fungal infections are a growing medical problem for patients after transplant and during immunosuppressive treatments. The associated healthcare costs of fungal infections are substantial [6]. Thus, learning more about fungal growth in various conditions could help find new ways to enhance or inhibit the growth of fungi, depending on the circumstances [2, 5–8].

Filamentous fungi can navigate and branch, making them well-adapted to spread in the environment, including their spreading on and within plant and animal tissues. Individual hyphae possess an ability to sense the physical and chemical properties of interfaces, and thus can respond to growth in different environments [9]. Branching is also crucial to the development of extensive fungal networks, facilitating nutrient acquisition. Overall, fungi display a great deal of diversity in growth patterns, which propagate from varying growth rates, hyphal dimensions, branching patterns, and frequency. Recently, the use of microfluidic devices has emerged as a powerful approach to studying how fungi grow in confined environments, relying on microscale models of environmental challenges, such as micron-sized topographies replicated in hard or soft polymers [10–13]. These models enable the study of fungal growth and branching under fluid perfusion [11]. They also facilitate understanding how fungi navigate around and through micron-sized barriers and channels, e.g., understanding the role of the fungal Spitzenkörper–microtubule complex in determining how hyphae navigate obstacles in their path [12]. A further advantage of microfluidic devices for fungal research is that they are compact and easily transportable, making them ideal substrates to use in collaborative research projects conducted with researchers across the globe [13].

The clear advantages of microfluidics for the measurement of hyphal growth rates and branching patterns are attracting more fungal research groups to use these technologies. However, where microfluidic devices have been used, current methodologies are often highly customized for each species and vary significantly between labs. In order to make microfluidics with live imaging easily accessible to research groups working on a range of fungi, we started a worldwide crowd-sourced collaborative effort, the Fungus Olympics. All participating labs were furnished with identical microfluidic devices and microscopy platforms. The event enabled the quantitative comparison of 14 strains representing nine species from three phyla of filamentous fungi across for growth rate, branching, and navigation strategies in confined spaces.

## Materials and methods

### Fungus Olympics logistics

The 2019 Fungus Olympics was organized by the Irimia and Momany labs. Participants for the contest were recruited via Twitter and email communication. There was no fee for entry. Winners were announced at the 2019 Mycological Society of America (MSA) conference.

## Fungus preparation (in alphabetical order)

*Aspergillus fumigatus-* frozen stock of conidia (AF293 isolate) was plated on Glucose Minimal Medium (GMM) plates and grown at 37˚C for 5–7 days by Shiv Kale. Conidia were harvested in PBS-Tween (0.1%) and immediately injected into the device at 5 μL of $1 \times 10^7$ cfu/mL in RPMI at 37˚C. The recording was immediately started with time intervals of 5 minutes.

*Aspergillus fumigatus-* ((A.f. *A. fumigatus* (A.f. 293.6 pyrG-/argB-) mKate2-rabA:argB; A.p. pyrG) or *A. fumigatus* Hook KO ((akuB- pyrG-) mKate2-rabA:hyg; Δ*hookA*:A.p. pyrG) conidia were harvested and counted in the Egan lab. Conidia were diluted to 5 x $10^7$ cfu/mL, and 0.5 μL was loaded into the device, primed with Glucose Minimal Media (1% Yeast Extract, 2% Glucose). The device was placed at 37˚C and imaged immediately with 5-minute intervals.

*Aspergillus fumigatus-* (invasive clinical isolate, accession number withheld for patient confidentiality) was cultured on potato dextrose agar and grown for 48–72 hours at 35˚C in the Coad lab. Conidia were suspended in saline using a drop of Tween 80 and diluted to $5 \times 10^7$ cfu/mL in Sabouraud's dextrose broth containing chloramphenicol and gentamicin. 10 μL of this solution was inoculated into the device, placed in the incubator at 35˚C, and recording started immediately. A time interval of 5 minutes was used.

*Aspergillus nidulans-* (Strains A850 (WT) and AYR32 (Δ*aspB*)) conidia were grown on solid complete media (1% glucose, 2% peptone, 1% yeast extract, 1% Casamino Acids, 0.01% vitamins and supplements, nitrate salt solution, and trace elements, pH 6.5); 1.8% agar was added for solid medium. Additional supplements were added depending on strains auxotrophic markers (i.e., pyridoxine HCl, p-aminobenzoate, riboflavin HCl, arginine, uridine, and uracil) in a dark incubator at 30˚C for approximately three days post-inoculation. Conidia were harvested in water, stored at 4˚C, and used within 30 days of harvesting. Spore inoculum was normalized to $2 \times 10^7$ cfu/mL in liquid complete-media, and 10 μL was loaded into the central loading chamber of the device. The microfluidics device was incubated at 30˚C for 12–16 hours post-inoculation or until the hyphae had grown into the obstacle before imaging. Imaging was conducted at 5 min time intervals in the 'zoomed-in' magnification. These *A. nidulans* entries from the Momany lab were used for protocol development and reference runs and were not a part of the competition.

*Cryptococcus neoformans-* (Strain KN99α, wild type reference strain) yeast cells were grown in liquid yeast extract-peptone-dextrose (YPD) supplemented with 2% glucose overnight at 30˚C and counted in the Nielsen lab [14]. Yeast cells were resuspended at $1 \times 10^7$ cfu/mL, and 5 μL was loaded into the device, primed with liquid YPD. The device was incubated for 14 hours at 30˚C and then imaged with 15-minute intervals.

*Magnaporthe oryzae-* (Strain B157, previously isolated from rice leaves) Wildtype (WT) and *dam1Δ* were grown on prune agar at 28˚C for 9 days (3 days in the dark followed by constant light) for conidiation in the Manjrekar lab [15]. The conidia were harvested in water and filtered through two layers of miracloth. WT conidia were inoculated into the device at 10 μL of $5 \times 10^5$ cfu/mL in Complete liquid media. This was grown at 28˚C for 3 or 24 hours before the start of recording. *M. oryzae* (Strain B157) *dam1Δ* conidia were inoculated into the device at 10 μL of $1 \times 10^7$ cfu/mL and grown for 24 hours at 28˚C before the start of timelapse recording. A time interval of 10 minutes was used.

*Metarhizium anisopliae* (ARSEF strain 549)- spores were harvested from the entire surface of a potato dextrose agar plate incubated at 22˚C in a dark incubator for one week. These spores were diluted to $1 \times 10^7$ cfu/mL in the St. Leger lab. 5 μL of this solution was loaded into the device, which was primed with SDB + 0.2% yeast extract media. The device was incubated for approximately 24 hours at 27˚C before being imaged with 15-minute intervals.

*Neurospora crassa-* spores were grown on Medium N without ammonium nitrate [16] with 1.5% agar slants at 22°C until conidiation by the Kronholm lab [16]. Two strains were used: C40 obtained from a cross, and 1131, a natural isolate, previously described in [17]. Conidia were suspended in 1 mL of 0.01% Tween-80, filtered, and counted. Conidia were diluted to $4.5 \times 10^6$ cfu/mL, and 2 μL was loaded into the device, which was primed and filled with Medium N. The device was incubated at 35°C for 10 hours (Strain C40) or 16 hours (Strain 1131) before imaging with 5-minute intervals. Starting concentration of conidia and time of incubation before starting imaging were determined by initial trials.

*Rhizopus microsporus-* (invasive clinical isolate, accession number withheld) was cultured on potato dextrose agar and grown for 24–48 hours at 35°C in the Coad lab. Conidia were suspended in saline and diluted to $5 \times 10^7$ cfu/mL in Sabouraud's dextrose broth containing chloramphenicol and gentamicin. 10 μL of this solution was inoculated into the device, placed in the incubator at 35°C, and recording started immediately. A time interval of 10 minutes was used.

*Rhizopus stolonifer-* (strain NRRL 66455) was grown for 1 week at 25°C on Malt Extract-Yeast Extract agar in the Stajich lab. *R. stolonifer* spores were suspended in 0.01% tween-80 [18]. The spores were counted and diluted to $1 \times 10^5$ cfu/mL. 10 μL were loaded into the device, which was primed and filled with liquid MEYE. The device was imaged at 5-minute intervals.

*Trametes versicolor-* Fruiting bodies of *Trametes versicolor* (WT) were harvested from a local tree stump in Oakland, CA. Stock samples were generated and stored as colonized PDA medium cubes suspended in 25% glycerol at -80°C. Species identity was confirmed by amplifying and sequencing the Internal Transcribed Spacer (ITS) region, using primers ITS1 and ITS4 [19]. Runs were always started from colonies actively growing on PDA plates. Before the runs, small fragments of PDA medium (about 1mm$^3$) colonized with actively growing mycelium (from the edge of the colony) were inoculated into 14 ml round-bottom Falcon tubes containing 1.5 ml YM media (Difco, #BD 271120) and grown at 30°C without shaking for 2–3 days until fluffy growth of submerged mycelia was visible. Mycelia in YM medium were disrupted into hyphal fragments by pipetting with a 1000 μl tip and then passing them repeatedly through a syringe with an 18 Gauge needle. Following the microfluidics device's priming, 10 μl of a suspension of hyphal fragments was introduced into the center of the device and incubated for 16–20 hours at 30°C. Old media and external fungal growth were then removed. A drop of fresh media was added to the center of the device, along with 2–3 ml of fresh media to the well. The recording was then initiated using a 15-minute time interval.

**Device fabrication.** Devices were designed using AutoCAD 2017 (v.O.48.M.294, Auto-Desk). Photolithography masks were printed by FineLine Imaging Inc (Colorado Springs, CO) and used to pattern silicon wafers with two layers of negative photoresist (SU-8, Microchem, Newton, MA). A 10 μm layer was used for the Olympic challenges, while large features such as the loading chamber were patterned on a 200 μm layer. Patterning was performed using sequential ultraviolet light exposure of the photoresist through respective photolithography masks and the wafers processed using standard microfabrication techniques according to the manufacturer's instructions. The wafer, patterned with 64 of the devices, was then used as a master mold for PDMS (Polydimethylsiloxane, Fisher Scientific, Fair Lawn, NJ) soft lithography. After curing, inlet holes were punched using a 1.5 mm punch (Harris Unicore), and each device released from the mold using a 5 mm outer punch. Devices were then irreversibly bonded to a 6 cm glass-bottom petri dish, following oxygen plasma treatment, with the bonding process being completed by placing the culture dishes on a hot plate at 85C for 10min.

**Microscopy setup.** Each of the participant labs employed a CytoSmart Lux2 microscope (Eindhoven, The Netherlands). The microscope was inserted into an incubator whenever the experiments required higher than room temperature. Automated time-lapse imaging was conducted in the 'Zoomed Out' or 'Zoomed In' settings (which correspond to 5X and 10X total

magnification, respectively) and one of three imaging time intervals (5, 10, and 15 minutes/image). These microscopes were connected to a portable laptop. Acquired microscopy images and temperature data were automatically sent to the CytoSmart cloud during the experiments. For the *R. stolonifer* follow-up experiments examining branching, an OMAX microscope with 4× magnification at 23˚C temperature was used with a 15-minute imaging interval.

**Device loading.** Devices and CytoSmart Lux2 microscopes were shipped out to each participating team before the competition. Fungal species were selected by each team, and a general loading protocol was shared with all participants (S1 File). This protocol served as a shared starting point for all groups, though they were encouraged to optimize it for their chosen fungi if necessary. Devices were primed by feeding an appropriate growth media through the central chamber until it visibly exited the ports on the side of the device. The central port was covered with a bubble of media and then placed in a vacuum chamber for 10 minutes. Following the vacuum, the device was allowed to equilibrate for 10 minutes, then screened by microscopy to ensure all features were filled with media. The media was added to fill the entire well and cover the device, approximately 3 mL. Fungi were loaded into the central port of the device and allowed to incubate at an optimal temperature for a certain time period (see individual materials for details on each species) to allow fungi to grow close to the features of the device. Time-lapse microscopy was then carried out on the provided CytoSmart microscopes to observe fungal growth through the device features.

**Data analysis.** Analysis of all time-lapse imaging datasets was performed manually. The analysis was performed for velocity and navigation and for as many features as possible for each entry. Several issues resulted in entries not being included in the analysis. While there was no time limit set on entries, if fungi did not reach at least halfway through the maze before the timelapse ended, the run was excluded. Fungal growth beginning within the maze, and inability to follow individual fungal hyphae through the features (overgrowth of the fungi, multiple hyphae entering simultaneously) were the other reasons for the exclusion of experiments from analysis. Individual hyphae were observed, with the time of entry and the time of exit out of the maze being used to determine 'time to escape' for each feature. For the three maze features, designated 'honeycomb,' 'square,' and 'boomerang,' all fungi that finished at least half the maze were analyzed, though only those that completed the whole maze were eligible for prizes. For determination of hyphal velocity, individual hyphae that grew in the "straight-line feature" were tracked manually in ImageJ (2.0.0-rc -59/1.52p) with Trackmate until growth ended or they left the field of view. For branching analysis, branching events were only counted for those that occurred on the leading hypha during its time within the maze as secondary hyphae entering the maze (or previous branches) could obscure events, not at the leading edge. For branch vs. nub determination, events of the leading edge were followed over time to observe growth. Branches went on to fully develop into long hyphae, while nubs arrested their growth and remained short (usually 1.5x the width of the parent hyphae or less).

**Statistics.** For statistical analyses and graphing, we used Prism (GraphPad Software version 8.3). Because sample sizes were small (N < 12), we employed the Kruskal-Wallis non-parametric test corrected using the false discovery rate method of Benjamini and Hochberg. Differences between means were considered significant at p<0.05. Values are represented on graphs as mean and Standard Deviation (SD).

## Results

We engaged the international fungal community to compare diverse filamentous fungal species for their ability to grow in microfluidic channels and navigate through microfluidic mazes. We sent the 'Fungus Olympics Devices' and portable benchtop CytoSmart microscopes

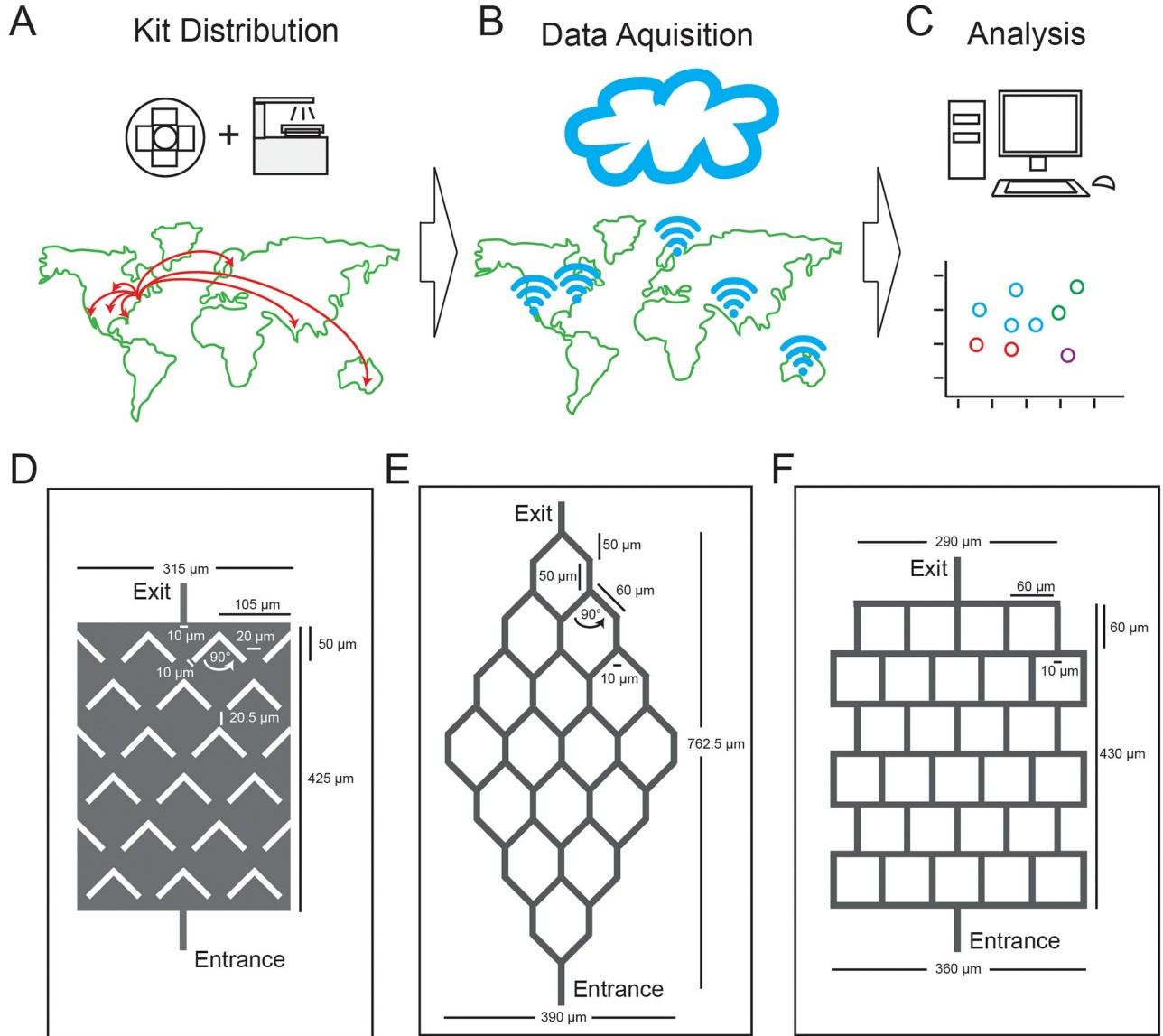

**Fig 1. The Fungus Olympics, an international collaborative study of fungal hyphae in microfluidic mazes.** A-C) Overview of approach: (A) Microfluidic devices and CytoSmart mini microscopes are distributed to participating research groups around the world; (B) Each group performs experiments and uploads results to an online server; (C) Data is analyzed and hyphal growth patterns compared. D-F) Microfluidic mazes for analyzing hyphal growth: (D) "Boomerang" maze contains a challenging array of angular obstacles designed to trap hyphae growing from the entrance at the bottom; (E) "Honeycomb" maze presents hyphae with hexagonal challenges requiring turns of 45˚; (F) "Square" maze presents hyphae with a series of 90˚ surfaces. Open growth channels are shown in gray. Three maze designs and straight channels for measuring hyphal velocity were present in each microfluidic device.

to fifteen labs in seven countries (Fig 1). The use of microfluidic devices and automated microscopes enabled us to gather consistent data from this substantial collaborative effort.

Our devices included straight-line, microscale channels for comparison of velocities of all fungal species. The velocity of fungi growing in straight channels was tracked manually, following the leading hyphal tip from frame to frame, as shown in Fig 2A. The organisms with the fastest average velocity were *Trametes versicolor*, *Aspergillus nidulans*, *Rhizopus stolonifer*, and *Neurospora crassa*. The overall results are presented in Table 1. Individual tracks

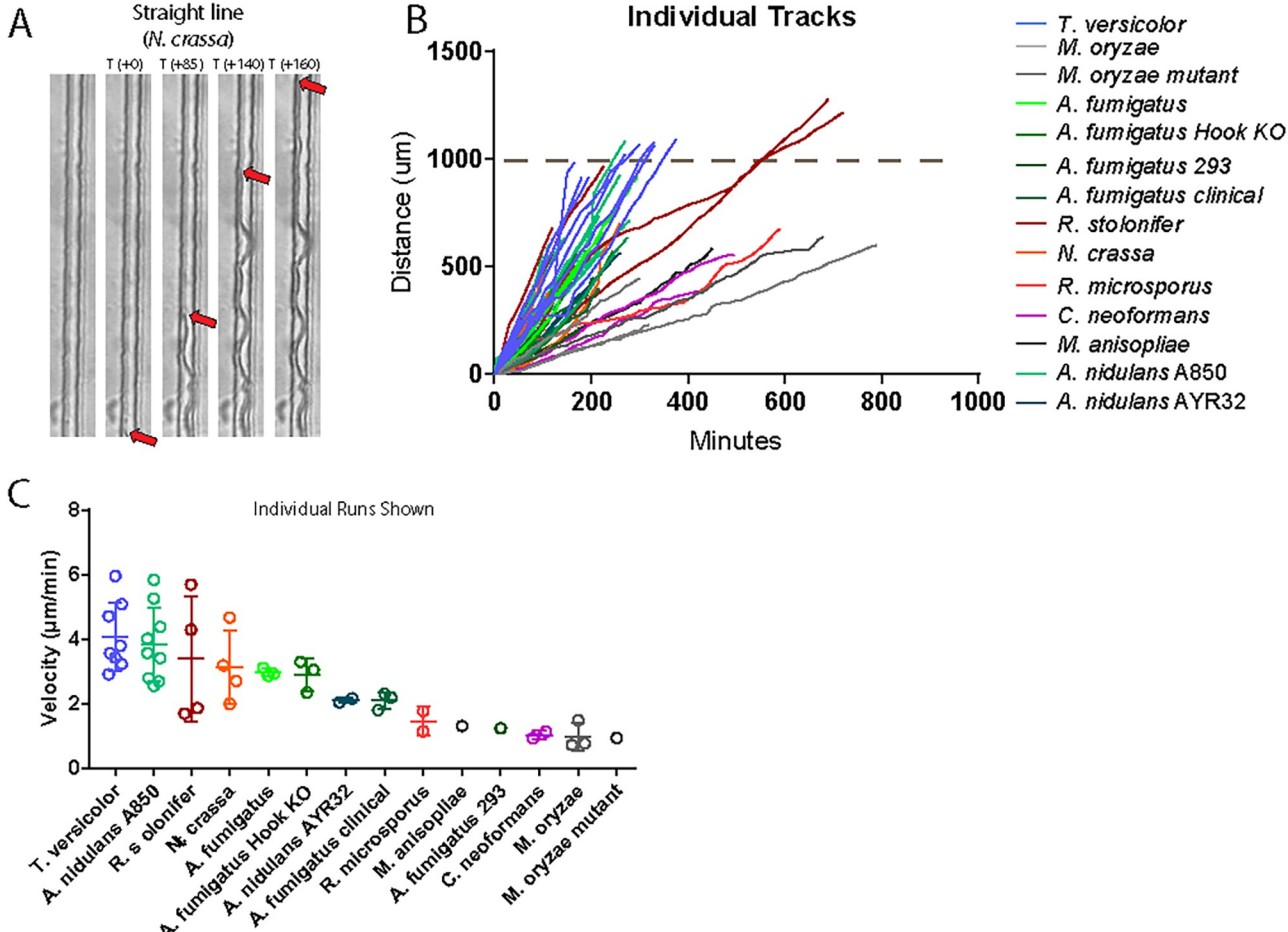

**Fig 2. Fungal velocities.** (A) A time series of a fungal hypha (*N. crassa*) growing in the straight channel. Red arrows indicate the leading hyphal tip, which was tracked for velocity determination. Channels are 10 μm wide. (B) Individual tracks for each fungus are shown, plotting the cumulative distance of hyphal growth (μm) vs. time (min). (C) The velocities are shown for each fungus. Individual runs are presented as open circles. Lines represent averages and standard deviation.

incorporating cumulative time and distance are shown (Fig 2B). The average velocities of the fungi in the Olympics varied from 6 (*T. versicolor*) to less than 1 μm/min (*M. oryzae*) (Fig 2C).

The 'Fungus Olympics Devices' also included three distinct mazes, in which channels or obstacles required growing hyphae to turn at defined angles (Fig 1). To navigate the honeycomb mazes, fungi had to execute 45-degree turns. To navigate the square mazes, fungi had to execute 90-degree turns. And to navigate the boomerang mazes, fungi had to execute turns of greater than 90 degrees. The boomerang maze also differs from the others because it consists of open space with embedded obstacles rather than channels. Examples of the three mazes are shown, with a time series from selected fungi showing typical maze navigation (Fig 3).

*T. versicolor* and *R. stolonifer* appeared to use very different growth strategies, with distinct advantages for navigating different maze angles (Fig 4). *R. stolonifer* appeared to explore most possible turns via branching, but *T. versicolor* was more selective, establishing one or more dominant branches and then proceeding past the turns on a distinct path (Fig 4 and S1–S6 Videos). Inside the honeycomb maze, *R. stolonifer* seemed to explore all options, but *T. versicolor* primarily occupied the outside channels of the maze, maintaining the same directional

**Table 1. Fungus Olympics challengers results and teams.**

| Species | Challenge | Speed (μm/min) | #Runs | Full run Time (hours) | Full run #Runs | Half run Time (hours) | Half run #Runs | Principal Investigator | Team | Location |
|---|---|---|---|---|---|---|---|---|---|---|
| *Trametes versicolor* | Square | | | 4.4 | 3 | | | David Breslauer | Paul Guerette, Edyta Szewczyk & Kevin McCluskey | Bolt Threads |
| | Honeycomb | | | 4 | 3 | | | | | |
| | Boomerang | | | 7 | 3 | | | | | |
| | Straight | 4.1 | 8 | | | | | | | |
| *Aspergillus nidulans* A850 | Square | | | 4.2 | 1 | 2.3 | 1 | Michelle Momany | Alex Mela | University of Georgia |
| | Honeycomb | | | | | 2.3 | 3 | | | |
| | Boomerang | | | | | | | | | |
| | Straight | 3.8 | 9 | | | | | | | |
| *Rhizopus stolonifer* (NRRL 66455) | Square | | | 5.2 | 2 | | | Jason Stajich | Derreck Carter-House & Jesus Pena | University of California, Riverside |
| | Honeycomb | | | 7.5 | 1 | 9.3 | 1 | | | |
| | Boomerang | | | 1.5 | 1 | | | | | |
| | Straight | 3.4 | 4 | | | | | | | |
| *Neurospora crassa* | Square | | | 4.5 | 1 | 1.9 | 1 | Ilkka Kronholm | Ilkka Kronholm | University of Jyväskylä |
| | Honeycomb | | | | | | | | | |
| | Boomerang | | | | | 1.4 | 1 | | | |
| | Straight | 3.1 | 4 | | | | | | | |
| *Aspergillus fumigatus* | Square | | | 5.3 | 1 | | | Martin Egan | Martin Egan and Baronger Bieger | University of Arkansas |
| | Honeycomb | | | 5.7 | 1 | 2.5 | 1 | | | |
| | Boomerang | | | | | | | | | |
| | Straight | 3 | 3 | | | | | | | |
| *Aspergillus fumigatus* Hook KO | Square | | | 3.9 | 1 | | | | | |
| | Honeycomb | | | | | 2.1 | 1 | | | |
| | Boomerang | | | 4.4 | 1 | | | | | |
| | Straight | 2.9 | 3 | | | | | | | |
| *A. nidulans* AYR32 | Square | | | | | | | Michelle Momany | Alex Mela | University of Georgia |
| | Honeycomb | | | | | | | | | |
| | Boomerang | | | | | | | | | |
| | Straight | 2.1 | 2 | | | | | | | |
| *Aspergillus fumigatus* (clinical isolate) | Square | | | | | | | Bryan Coad | Bryan Coad and Sarah Kidd | The University of Adelaide |
| | Honeycomb | | | 6.8 | 1 | | | | | |
| | Boomerang | | | | | | | | | |
| | Straight | 2.1 | 3 | | | | | | | |
| *Rhizopus microsporus* (clinical isolate) | Square | | | | | | | | | |
| | Honeycomb | | | | | 10 | 1 | | | |
| | Boomerang | | | | | 5.5 | 1 | | | |
| | Straight | 1.5 | 2 | | | | | | | |
| *Metarhizium anisopliae* | Square | | | | | 5.8 | 1 | Raymond St. Leger | Brian Lovett and Huiyu Sheng | University of Maryland, College Park |
| | Honeycomb | | | | | 6.2 | 1 | | | |
| | Boomerang | | | | | | | | | |
| | Straight | 1.3 | 1 | | | | | | | |
| *Aspergillus fumigatus* 293 | Square | | | | | | | Shiv Kale | Shiv Kale | Nutritional Immunology and Molecular Medicine Institute |
| | Honeycomb | | | | | 3.7 | 1 | | | |
| | Boomerang | | | | | | | | | |
| | Straight | 1.2 | 1 | | | | | | | |

(*Continued*)

**Table 1.** (Continued)

| Species | Challenge | Speed (µm/min) | #Runs | Full run Time (hours) | #Runs | Half run Time (hours) | #Runs | Principal Investigator | Team | Location |
|---|---|---|---|---|---|---|---|---|---|---|
| *Cryptococcus neoformans* (K99alpha) | Square | | | | | | | Kirsten Nielsen | Sophie Altamirano | University of Minnesota |
| | Honeycomb | | | | | | | | | |
| | Boomerang | | | | | | | | | |
| | Straight | 1 | 2 | | | | | | | |
| *Magnaporthe oryzae* (B157) | Square | | | | | | | Johannes Manjrekar | Hiral Shah | The Maharaja Sayajirao University of Baroda |
| | Honeycomb | | | | | | | | | |
| | Boomerang | | | | | | | | | |
| | Straight | 0.98 | 3 | | | | | | | |
| *Magnaporthe oryzae* dam1Δ mutant | Square | | | | | | | | | |
| | Honeycomb | | | | | | | | | |
| | Boomerang | | | | | | | | | |
| | Straight | 0.94 | 1 | | | | | | | |

Linear growth speed and time to cross microfluidic devices with four different designs for the fourteen filamentous fungi in the Fungus Olympics. The fastest growing fungi are listed first. Principal investigators, teams, and locations are also listed.

growth (S1 and S2 Videos). Inside the square maze, at +90 min, two rows in, *T. versicolor* only had two leading hyphae exploring, while *R. stolonifer* (at +210 min), at the same position in the maze, had continued to grow through the top row and occupied many of the turns that *T. versicolor* ignored (S3 and S4 Videos). Inside the boomerang maze, (at +30 min) *T. versicolor* had one hypha that had hit the first obstacle, whereas at +15 min, *R. stolonifer* had already explored the top row via branching (S5 and S6 Videos). Examining the number of branches vs. time to escape clearly shows that branching slows navigation of the honeycomb and square mazes but speeds navigation of the boomerang maze (Figs 4 and 5). The *R. stolonifer* strain had a higher rate of branching in all mazes, and this appeared to give an advantage over other species in the boomerang maze (Fig 5C and 5D).

For each fungus that fully completed a maze, 'time to escape' was determined by following the leading hyphal tip from the time it entered the maze to the time it exited (Fig 5). Fungi with the highest linear velocities were not always the fastest in navigating mazes. The time to escape varied depending on the maze pattern (Fig 5C). Two organisms were particularly interesting. Relative to other fungi we analyzed, *T. versicolor* was fastest in straight-line velocity and in escape from the honeycomb maze and one of the fastest in escape from the square maze. However, *T. versicolor* was much slower to escape from the boomerang maze (Fig 5C). The opposite was true of *R. stolonifer*, which was intermediate in straight-line velocity, slowest to escape from the honeycomb and square mazes, but quickest to escape the boomerang maze (Fig 5C).

To better investigate the branching patterns of *R. stolonifer*, we repeated straight-line velocity and maze escape experiments at higher magnifications. We tracked the speed of individual hyphae and compared the original and new runs (Fig 6). The average velocities for all hyphae in straight channels were comparable (p = 0.78, N = 4 individual hyphae across 2 experimental runs for the "old" data from the competition; N = 9 individual hyphae across 5 experimental runs for the "new" post-competition data). The average time to escape (full runs only) for each maze was comparable between old and new experiments. The new set of experiments showed that in addition to many branches, *R. stolonifer* formed small protrusions of less than 5 µm

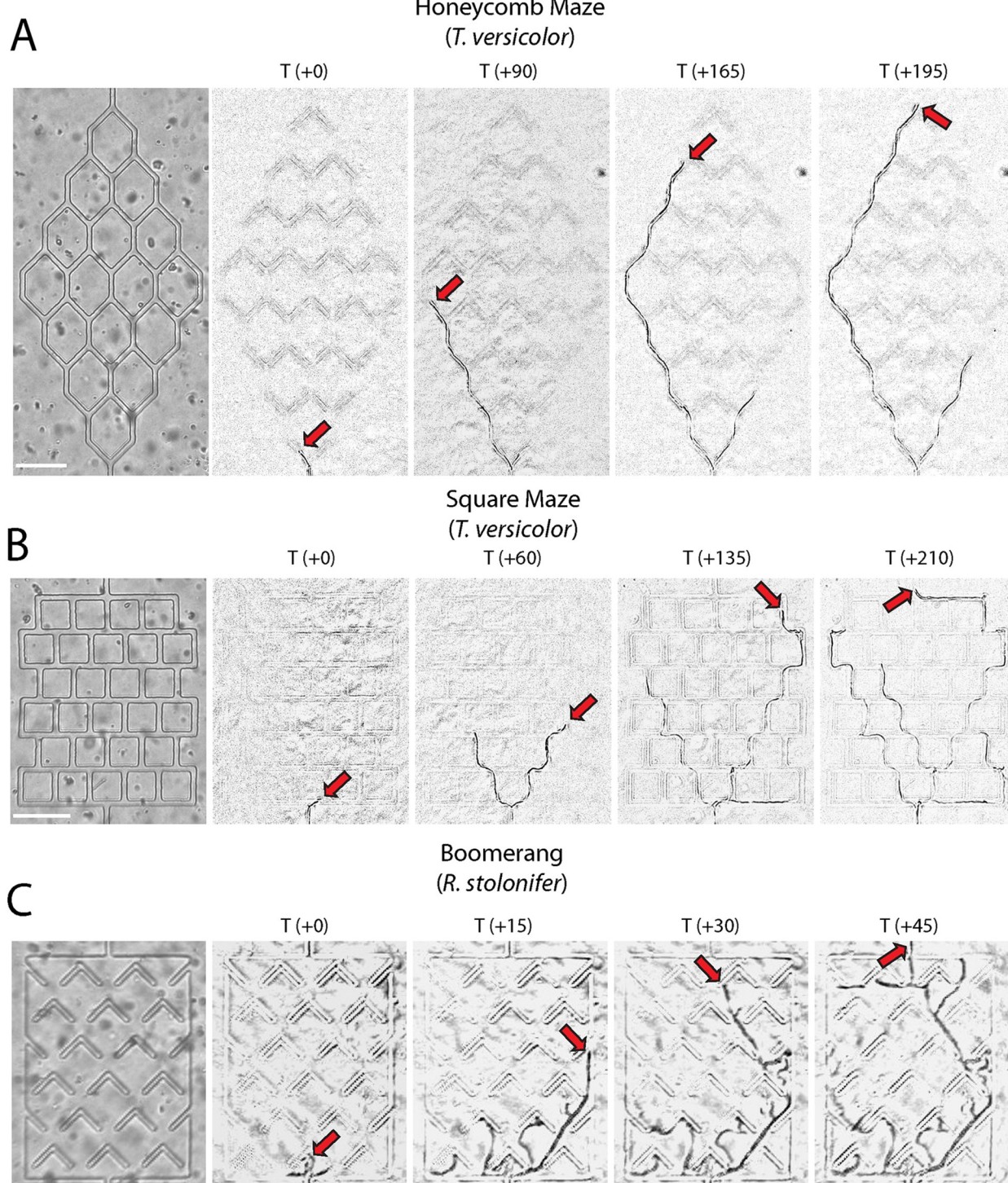

**Fig 3. Fungal growth in maze obstacles.** (A) Honeycomb mazes challenge the growing hyphae with channel bifurcations at acute angles. (B) Square mazes consist of orthogonal channels forcing right-angle turns for growing hyphae. (C) Boomerang obstacles are such that growing hyphae have to turn more than 90 degrees to avoid being trapped. Red arrows indicate the leading hyphal tip. (T) is time, in minutes, from the point the hypha enters the channel. Scale bars are 100 μm.

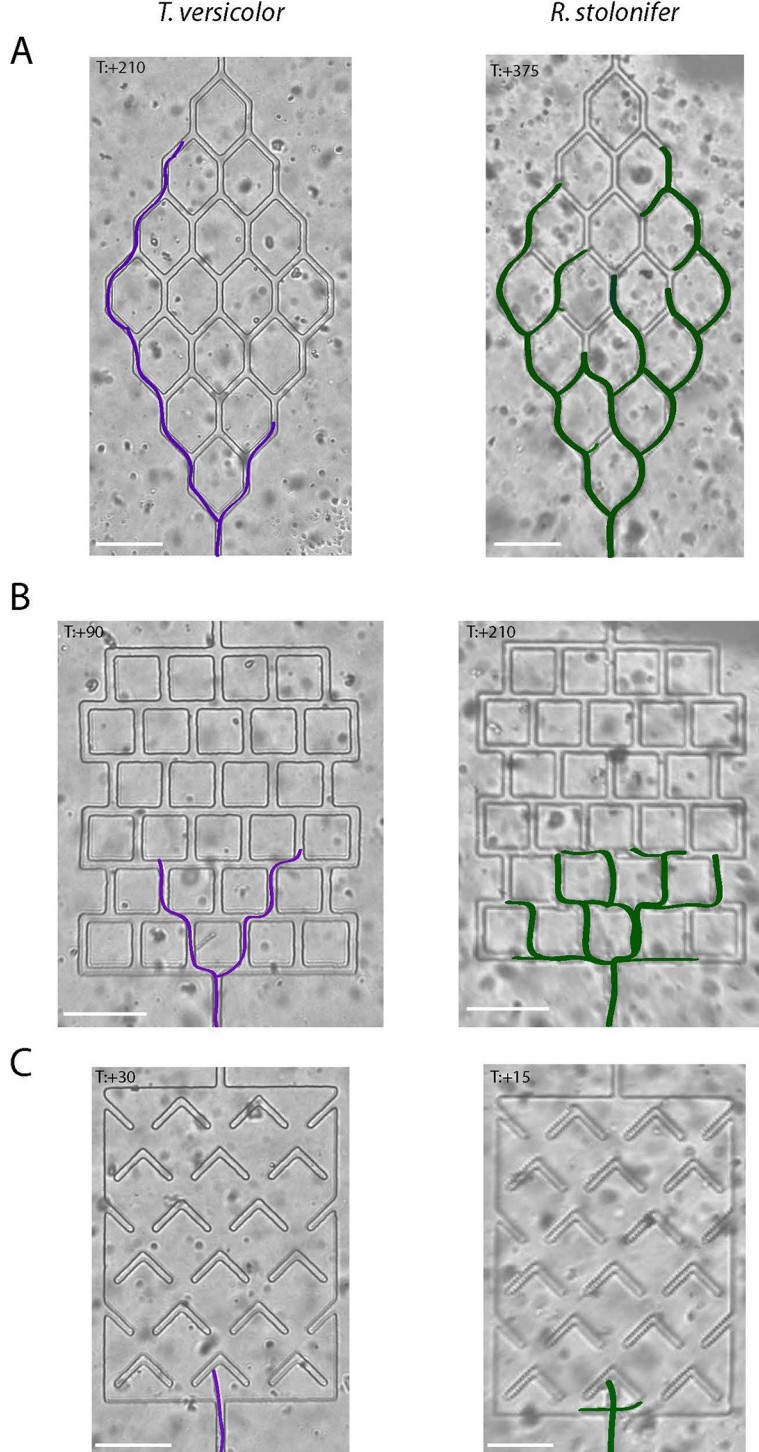

**Fig 4. Growth strategies.** *T. versicolor* and *R. stolonifer* use different growth strategies for navigating different mazes. *T. versicolor* establishes a dominant branch that grows faster. *R. stolonifer* branches often; however, each branch grows slower. *T. versicolor* strategy appears to be more efficient in devices with simple challenges, like the bifurcating channels. *R. stolonifer* branching strategy is an advantage when confronting complex challenges, like the boomerang traps. Scale bars are 100 μm. To make growth patterns more easily visible, *T. versicolor* hyphae were traced in purple and *R. stolonifer* hyphae were traced in green.

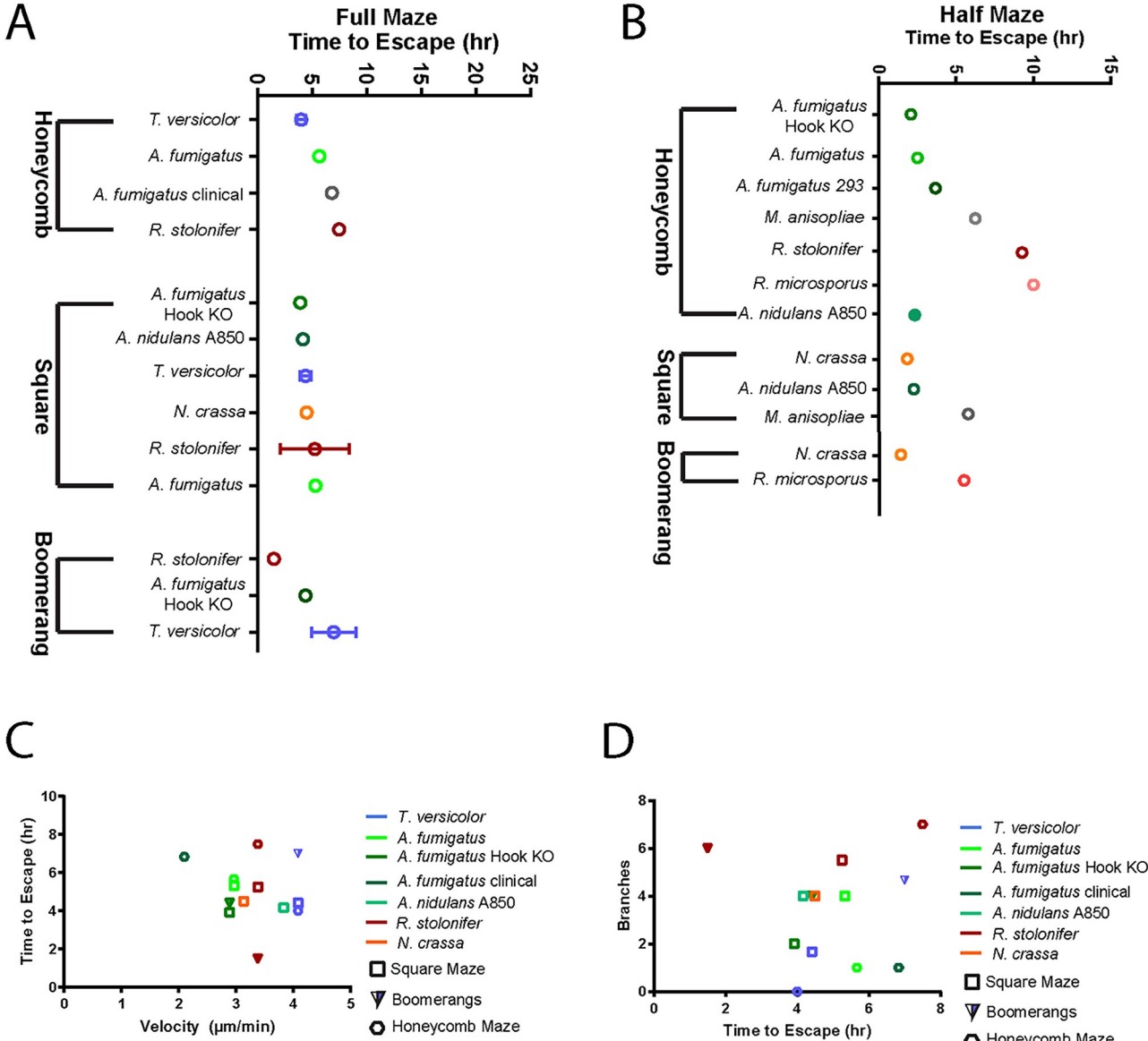

**Fig 5. Maze escape analysis.** The average time it took to escape each maze is displayed, broken down by individual fungi and maze type. (A) The data for fungi that completed entire mazes is shown. (B) Data for fungi that did not complete the full mazes but completed at least half are also shown. (C) Fungi that completed entire mazes are plotted comparing average time to escape vs. average straight-line velocity (from Fig 2). (D) Average branch number vs. average time to escape. Error bars represent standard deviation.

that we called 'nubs' (Fig 7). Live imaging showed that these protrusions did not extend over time. We found that for *R. stolonifer* in all three mazes, the number of nubs increased with decreasing hyphal velocity. In contrast, the number of branches on each hypha was independent of the hyphal velocity (Fig 7B). Moreover, the number of branches and nubs and the time to escape appeared to be independent (Fig 7C). Though the role of nubs is not clear, it seems possible that they might be branch initials that fail to extend when hyphal velocity slows.

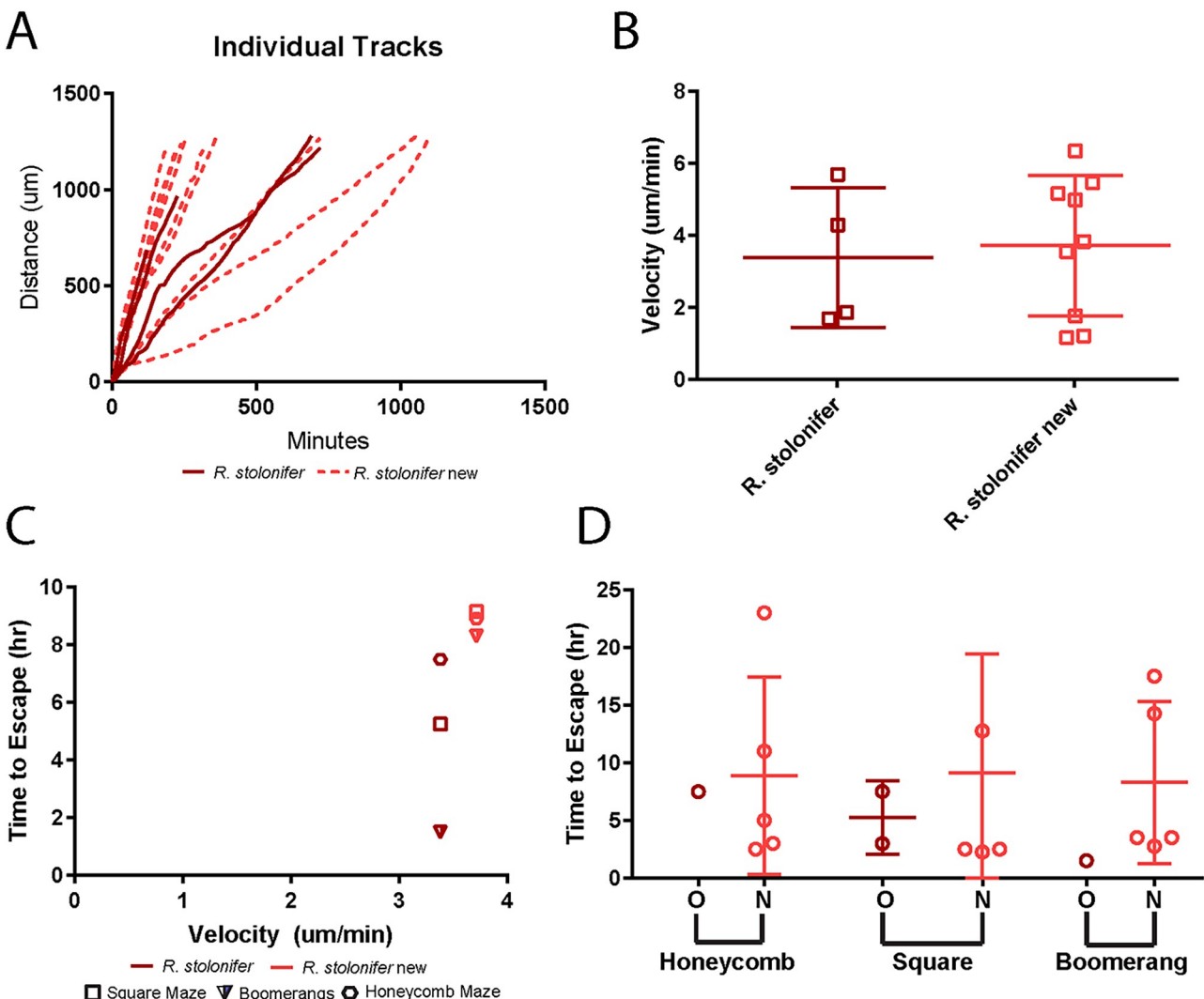

**Fig 6. Reproducibility of competition results.** After the initial runs by all labs, a second set of *R. stolonifer* experiments were conducted to establish reproducibility. (A) Individual hyphal tracks are shown for both the original runs (solid lines) and the new runs (dashed lines). (B) The average velocities were quantified. Individual dots represent the average velocity for each unique hypha in a straight channel. Error bars represent the mean ± standard deviation. (C) The average time to escape (full runs only) for each maze type is shown and plotted against the average velocity for the old and new runs. (D) The time to escape each maze is broken down to show individual maze data for old and new experiments. N = 4 hyphae, N = 2 experimental runs during the competition; N = 9 hyphae, N = 5 runs for the lab re-run for velocity data (A-B).

## Discussion

In this work, we ran a crowd-sourced collaborative project involving fifteen laboratories that used microfluidic devices and emerging imaging technologies to collect data on the growth of fourteen fungi. Our project is the first of this kind in the fungal field and was inspired by similar competitions among bacteria [20], cancer cells [21], and amoeba and neutrophils [22]. The labs used identical microscope systems and settings. The microfluidic devices provided were similar for all groups and were set to upload time-lapse movies to a cloud repository automatically. Participating teams had significant autonomy in choosing the fungal species, strains and growth conditions in order to optimize the performance of their entry. After the event, we

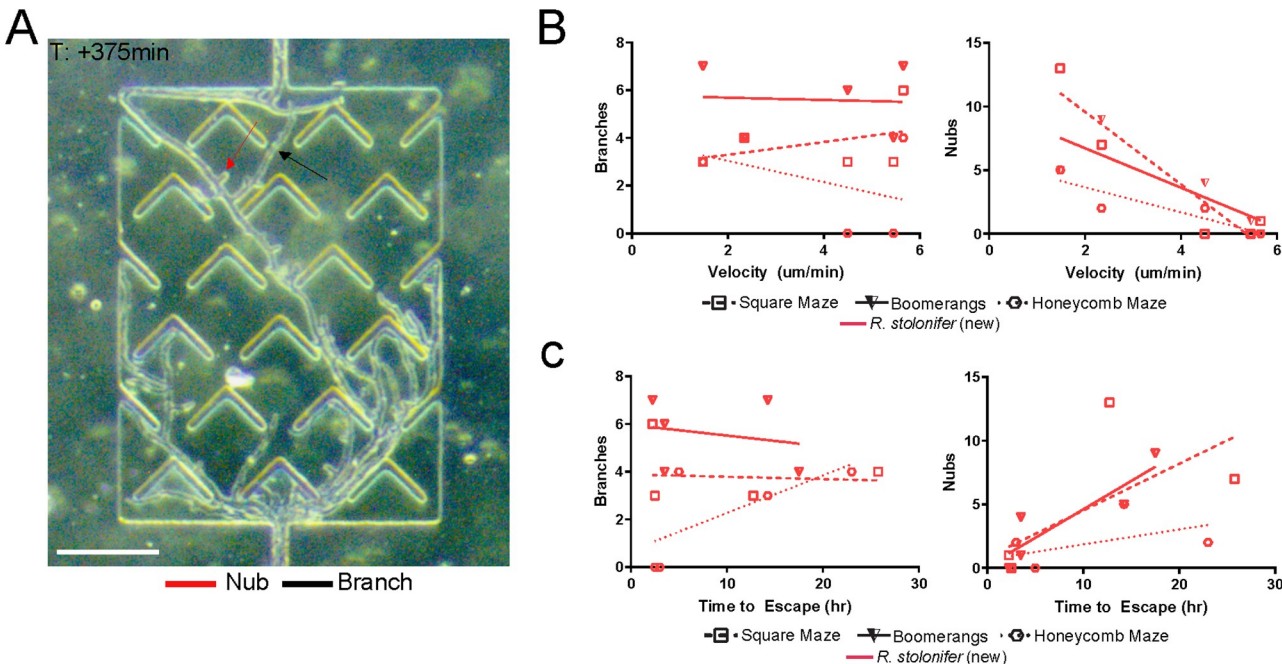

**Fig 7. Branching and nubbing of growing *R. stolonifer* hyphae.** *R. stolonifer* displayed a unique phenotype during hyphal growth, sometimes forming fully developed branches, other times creating small branches that did not grow further, termed "nubs" here (see methods for full criteria). (A) A full branch and a hyphal nub are identified in the image. Scale bar is 100 μm. (B) We found that the number of branches on each hypha is independent of the hyphal velocity. The number of nubs increases with decreasing hyphal velocity. (C) We found no relationship between the time to escape and the number of branches and nubs.

analyzed growth and navigation metrics using movies uploaded to the cloud during the experiments by all teams. We found that the three out of 14 organisms that performed the best in terms of average velocity were *T. versicolor* [23], *R. stolonifer* [24], and *N. crassa* [25]. Only two organisms, *T. versicolor* (a basidiomycete) and *R. stolonifer* (a zygomycete), completed all three mazes. Interestingly, when we compared mutant strains to WT counterparts, we consistently found mutants to perform less effectively than the wild type strains.

We measured hyphal growth velocity in microfluidic devices that focused the fungal growth into micron-scale channels in a straight line. The organisms that showed the highest average velocity, *T. versicolor*, *R. stolonifer*, *and N. crassa*, are all saprotrophs that grow on dead plants and represent three phyla: Basidiomycota, Mucoromycota, and Ascomycota, respectively. *T. versicolor* is a polypore that grows on dead wood, whereas *R. stolonifer* grows on rotting fruit [26] and *N. crassa* on burned vegetation [27]. It is surprising that *T. versicolor* grew the fastest out of these species, as polypores are often slow-growing. In contrast, fungi from the genus *Rhizopus* often grow in resource-rich, but ephemeral habitats, and many species are capable of only utilizing simple sugars. *Neurospora* is quite often found in more ephemeral habitats as well, which require fast growth for quick monopolization of resources. Earlier reports of growth rate on agar plates or tubes place *T. versicolor* as the slowest growing species of these [28, 29]. Growing in liquid in a small cavity seemed to slow the growth of *T. versicolor* the least. Species that exhibited a lower average velocity were primarily opportunistic or obligate pathogens of living organisms: *M. anisopliae* (insects) [30], *C. neoformans* (mammals) [31], *R. microsporus* (humans/animals/plants) [32], and *M. oryzae* (plants) [33, 34]. The exception to this finding is the opportunistic pathogen of humans, *A. fumigatus*, which grew rapidly. It

should be noted that though the pathogen *C. neoformans* is capable of hyphal growth, it reproduces asexually through budding [35]. Given that it was grown under conditions that favor yeast phase growth during the Fungus Olympics, it is not surprising *C. neoformans* yeast cells exhibited a lower average velocity compared to the filamentous fungi in this competition. [36].

We noted that none of the mutant strains measured performed as well as their wild-type counterparts. Such mutants included an *A. nidulans* septin mutant, Δ*aspB* (AYR32), an *A. fumigatus* mutant perturbed in dynein-mediated early endosome trafficking, Hook KO (Δ*hookA*), and the *M. oryzae* Δ*dam1* mutant. Dam1 is an outer kinetochore, a microtubule-associated protein that localizes to the growing hyphal tip, and its loss impairs hyphal morphology and branching. Future trials, including a broader range of participating labs, organisms, and gene-deletion mutants, could help further elucidate interesting growth patterns by species and ecological niches. These experiments could eventually lead to a better understanding of the mechanisms governing hyphal navigation and steering in filamentous fungi.

Though many of the fungi in the competition completed individual mazes, only *T. versicolor* and *R. stolonifer* completed all three mazes. Though we did not set out to examine hyphal branching and exploration strategies, something that has been the subject of significant study by other groups, the differences in exploration strategies of *T. versicolor* and *R. stolonifer* were striking [37–39]. *T. versicolor* established one or two dominant hyphae and did not branch into other turns. In contrast, *R. stolonifer* frequently branched, growing into almost every possible turn. With the more selective strategy, *T. versicolor* traversed mazes requiring 45 degrees (honeycomb) or 90 degrees (square) turns more rapidly than *R. stolonifer*. On the other hand, with the frequent branching strategy, *R. stolonifer* traversed the maze requiring turns greater than 90 degrees (boomerang) more rapidly. Our experiments did not find a significant relationship between the time to escape and the number of branches or nubs formed by *R. stolonifer*. Future experiments specifically designed to probe this question may shed more light on this relationship in the future. We did, however, note that *R. stolonifer* started more branches in the boomerang maze than in the honeycomb or square mazes. These observations raise the intriguing possibility that *R. stolonifer* evasive branching might be a default response when encountering mechanical obstacles, similar to earlier observations in *A. fumigatus* [40]. Alternatively, branching might be suppressed in spatially constricted channels like those found in the honeycomb and square mazes instead of more open areas in the boomerang maze. It is also possible that differences in nutrient availability and metabolism might have contributed to different growth patterns, as has recently been shown at the colony level for several grassland saprotrophic fungi [41].

Several laboratories focused on the clinically relevant and opportunistic pathogen *A. fumigatus* entered this competition [42]. Current and past research has pointed to the importance of isolate heterogeneity for germination [43], low oxygen fitness [44], heterogeneity of fungal surface [45], elicitation of an immune response [46], and nutrient acquisition [47], as contributors to pathogenicity. In this first Olympiad, *A. nidulans* isolates exhibited a higher average velocity than the *A. fumigatus* isolates. However, this may be attributed to the low number of successfully observed replicates achieved for the *A. fumigatus* isolates. Runs involving *A. nidulans* and other fungal isolates with many replicates indicated instances of both slow and fast velocities for a given run. In fact, as the average velocity increased for a given isolate, so did the variance. This preliminary finding may suggest heterogeneity in velocity within an isolate (some hyphae being tortoises and some being hares), or illuminate the natural variation in growth velocity for a given hypha at a given moment in time. Interestingly, clinical isolates examined in this study exhibited an average velocity similar to the other opportunistic fungal pathogens.

Overall, the first edition of the Fungus Olympics revealed differences in linear velocities and growth patterns. To our surprise, the analysis of time to traverse mazes revealed two distinct branching strategies with advantages for different conformations. High-frequency branching in which all possible paths were explored allowed faster escape from mazes featuring turns greater than 90 degrees (boomerang). Low-frequency branching in which only one or two paths were explored allowed faster escape from mazes featuring turns less than 90 degrees (honeycomb and square). These experiments demonstrate the utility of standardized platforms for comparing experimental results of a wide array of fungi from laboratories across the world. Future editions of the Fungus Olympics will allow us to explore further growth modes for a more extensive selection of wild-type and mutant fungal species.

## Supporting information

**S1 File. General protocol: The general protocol for loading fungi into the microfluidic device that was sent to all participating groups in the Fungal Olympics is shown here.**
(DOCX)

**S1 Data. Source data: Source data for the information summarized in Figs 2b, 2c, 5a, 5d, 6a, 6d, 7b and 7c.**
(XLSX)

**S1 Video. Timelapse of *T. versicolor* in honeycomb maze.** Time begins (T0) on the frame where the hyphae enters the maze, in minutes.
(AVI)

**S2 Video. Timelapse of *R. stolonifer* in honeycomb maze.** Time begins (T0) on the frame where the hyphae enters the maze, in minutes.
(AVI)

**S3 Video. Timelapse of *T. versicolor* in square maze.** Time begins (T0) on the frame where the hyphae enters the maze, in minutes.
(AVI)

**S4 Video. Timelapse of *R. stolonifer* in square maze.** Time begins (T0) on the frame where the hyphae enters the maze, in minutes.
(AVI)

**S5 Video. Timelapse of *T. versicolor* in boomerang maze.** Time begins (T0) on the frame where the hyphae enters the maze, in minutes.
(AVI)

**S6 Video. Timelapse of *R. stolonifer* in boomerang maze.** Time begins (T0) on the frame where the hyphae enters the maze, in minutes.
(AVI)

## Acknowledgments

We would like to thank all the labs who participated in the Fungus Olympics:

- Dr. Bryan Coad (The University of Adelaide) and Dr. Sarah Kidd (National Mycology Reference Centre, South Australia Pathology). Australia.

- Hiral Shah in the lab of Johannes Manjrekar at Bharat Chattoo Genome Research Centre, Dept. of Microbiology and Biotechnology Centre, The Maharaja Sayajirao University of Baroda, India.

- Sophie Altamirano in the lab of Kirsten Nielsen at the University of Minnesota, MN, USA.

- Ilkka Kronholm at the University of Jyväskylä, Finland

- Huiyu Sheng and Brian Lovett in the lab of Raymond St. Leger at the University of Maryland, College Park, MD, USA.

- Derreck Carter-House and Jesús Peña in the lab of Jason Stajich at the University of California, Riverside, CA, USA.

- Shiv Kale at the Nutritional Immunology and Molecular Medicine Institute, VA, USA.

- Baronger Bieger in the lab of Martin Egan at the University of Arkansas, AR, USA.

- Alex Mela in the lab of Michelle Momany at the University of Georgia, GA, USA

- David Peris Navarro and Carla Perpiñá at the Institute of Agrochemistry and Food Technology, Spain.

- Alex Andrianopoulos at the University of Melbourne, Australia.

- Iuliana Ene and Chapman Beekman in the lab of Richard Bennett at Brown University, RI, USA.

- Daniel Henk at the University of Bath, UK.

- Meritxell Riquelme at Centro de Investigación Científica y de Educación Superior de Ensenada CICESE, Mexico.

- David Breslauer, Paul Guerette, Edyta Szewczyk and Kevin McCluskey in the labs of Bolt Threads Inc., Emeryville, CA, USA

The Fungus Olympics idea, basic organization, and outreach to the fungal community were from Michelle Momany and Daniel Irimia. Fungus Olympics mazes were designed, and prototypes were fabricated by Dr. Felix Ellett at the BioMEMS Center. The devices used by the participants were fabricated and shipped by Anika Marand. The website for the event was set up by Andreu Cullere. Dr. Alex Hopke scheduled and organized the event and analyzed the data. Dr. Alex Mela contributed to experimental design and optimization. Alex Hopke, Alex Mela, Felix Ellett, Michelle Momany, and Daniel Irimia wrote the manuscript with input from all the authors.

We would also like to acknowledge the Mycology Society of America (https://msafungi.org/) for allocating time to present the results of the Fungus Olympics to the fungal community during the 2019 MSA conference.

## Author Contributions

**Conceptualization:** Alex Hopke, Alex Mela, Felix Ellett, Michelle Momany, Daniel Irimia.

**Formal analysis:** Alex Hopke.

**Funding acquisition:** Michelle Momany, Daniel Irimia.

**Investigation:** Alex Mela, Felix Ellett, Jesús F. Peña, Jason E. Stajich, Sophie Altamirano, Brian Lovett, Martin Egan, Shiv Kale, Ilkka Kronholm, Paul Guerette, Edyta Szewczyk, Kevin McCluskey, David Breslauer, Hiral Shah, Bryan R. Coad, Michelle Momany.

**Methodology:** Alex Mela, Felix Ellett, Derreck Carter-House, Jesús F. Peña, Michelle Momany.

**Project administration:** Alex Hopke.

**Resources:** Daniel Irimia.

**Writing – original draft:** Alex Hopke, Alex Mela, Felix Ellett, Derreck Carter-House, Jesús F. Peña, Jason E. Stajich, Sophie Altamirano, Brian Lovett, Martin Egan, Shiv Kale, Ilkka Kronholm, Paul Guerette, Edyta Szewczyk, Kevin McCluskey, David Breslauer, Hiral Shah, Bryan R. Coad, Michelle Momany, Daniel Irimia.

**Writing – review & editing:** Alex Hopke, Alex Mela, Felix Ellett, Derreck Carter-House, Jesús F. Peña, Jason E. Stajich, Sophie Altamirano, Brian Lovett, Martin Egan, Shiv Kale, Ilkka Kronholm, Paul Guerette, Edyta Szewczyk, Kevin McCluskey, David Breslauer, Hiral Shah, Bryan R. Coad, Michelle Momany, Daniel Irimia.

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
