## [Decision Letter · Decision Letter 0]

8 Apr 2021

PONE-D-21-05622

Crowdsourced Analysis of Fungal Growth and Branching on Microfluidic Platforms

PLOS ONE

Dear Dr. Irimia,

Thank you for submitting your manuscript to PLOS ONE. After careful consideration, we feel that it has merit but does not fully meet PLOS ONE’s publication criteria as it currently stands. Therefore, we invite you to submit a revised version of the manuscript that addresses the points raised during the review process.

This was an interesting, multidisciplinary study, with important implications for understanding fungal growth dynamics, that was generally well received by the reviewers. However, both reviewers had suggestions for improvement. Reviewer 1 and 2 requested more experimental details and for inconsistencies in study to be addressed. Reviewer 2 also noted a fundamental problem, namely that it was unclear what type of manuscript this is purporting to be: a "how to" protocol or a study of fungal growth patterns. To resolve this issue, as indicated by Reviewer 2, more recourse to previous primary literature is likely warranted.

We look forward to receiving your revised manuscript.

Kind regards,

Richard A Wilson

Academic Editor

PLOS ONE

Journal Requirements:

Please provide the accession numbers for the clinical isolates used in your study. If these are withheld at the request of a data access committee or ethics committee, please clarify this in your ethics and/or data availability statements as appropriate.

Thank you for stating the following in the Competing Interests section:

The authors have declared that no competing interests exist.

We note that one or more of the authors are employed by a commercial company: Bolt Threads Inc.,

3a) Please provide an amended Funding Statement declaring this commercial affiliation, as well as a statement regarding the Role of Funders in your study. If the funding organization did not play a role in the study design, data collection and analysis, decision to publish, or preparation of the manuscript and only provided financial support in the form of authors' salaries and/or research materials, please review your statements relating to the author contributions, and ensure you have specifically and accurately indicated the role(s) that these authors had in your study. You can update author roles in the Author Contributions section of the online submission form.

3b) Please also provide an updated Competing Interests Statement declaring this commercial affiliation along with any other relevant declarations relating to employment, consultancy, patents, products in development, or marketed products, etc. 

We note that Figure 1 in your submission contain map images which may be copyrighted. All PLOS content is published under the Creative Commons Attribution License (CC BY 4.0), which means that the manuscript, images, and Supporting Information files will be freely available online, and any third party is permitted to access, download, copy, distribute, and use these materials in any way, even commercially, with proper attribution. For these reasons, we cannot publish previously copyrighted maps or satellite images created using proprietary data, such as Google software (Google Maps, Street View, and Earth). For more information, see our copyright guidelines: http://journals.plos.org/plosone/s/licenses-and-copyright.

4a, You may seek permission from the original copyright holder of Figure 1 to publish the content specifically under the CC BY 4.0 license. 

4b, If you are unable to obtain permission from the original copyright holder to publish these figures under the CC BY 4.0 license or if the copyright holder’s requirements are incompatible with the CC BY 4.0 license, please either i) remove the figure or ii) supply a replacement figure that complies with the CC BY 4.0 license. Please check copyright information on all replacement figures and update the figure caption with source information. If applicable, please specify in the figure caption text when a figure is similar but not identical to the original image and is therefore for illustrative purposes only.

Reviewers' comments:

Reviewer's Responses to Questions

**Comments to the Author**

1. Is the manuscript technically sound, and do the data support the conclusions?

Reviewer #1: Yes

Reviewer #2: Yes

2. Has the statistical analysis been performed appropriately and rigorously? 

Reviewer #1: Yes

Reviewer #2: Yes

3. Have the authors made all data underlying the findings in their manuscript fully available?

Reviewer #1: Yes

Reviewer #2: Yes

4. Is the manuscript presented in an intelligible fashion and written in standard English?

Reviewer #1: Yes

Reviewer #2: Yes

5. Review Comments to the Author

Reviewer #1: Summary

First, I would like to thank the authors for organising the “Fungus Olympics” and their contribution to facilitating global collaboration efforts.

In this study, the authors present new device to characterise fungal growth velocity and branching frequencies and demonstrate its utility in gathering data from ascomycetes, basidiomycetes, and zygomycetes. They discovered two distinct strategies that fungi use to complete mazes. High branching frequencies allowed fungi to complete mazes with sharp turns faster, while low branching frequencies lead to an advantage navigating mazes with shallower turns.

While the authors present a compelling and coherent manuscript, addressing the missing data in a bit more detail in the material and methods section, as well as discussing the potential limitations of the study regarding the comparability of the different data sets, could further improve the quality of this manuscript. Otherwise, I am looking forward to reading about the second round of the “Fungus Olympics”.

Examples and evidence

Minor issues and suggestions.

While the authors are very thorough in describing which lab did which experiments, there are some minor points in the materials and methods section that need to be clarified for better understanding and reproducibility.

1. The authors use different units [c/mL (97,101,106,117, 152, 143, 149), cfu/mL (132, 134), spore/mL (137, 153)] to describe the concentration of the spore solution used to load the mazes. Can these be converted, so the same unit is used throughout the paper?

2. The culture conditions are described in great detail for most fungi. However, the culture conditions of Metarhizium anisopliae (136 - 139) to produce spores could be more detailed for better understanding. The authors should also mention how long Rhizopus stolonifera (152 – 154) was incubated before spores were harvested.

3. The authors do not mention which team, or teams, were conducting the experiments on Neurospora crassa (140 – 146). This information can be found in Table 1, but all the other contributing lab groups were mentioned in the material and methods section.

4. The authors describe criteria for excluding some entries from the analysis (212 – 214). One of them is insufficient fungal growth (less than half the maze). They do not specify whether there was a time limit for completing at least half of the maze. Clarifying the criteria that disqualified fungi would improve the overall understanding of the data analysis section.

5. The growth conditions, such as media and temperature, vary quite a bit between the entries. The authors should add a couple of sentences explaining how those conditions were chosen. Are they the ideal growth conditions determined by each participating lab group? The impact the varying culture conditions might have on the comparability of results should be briefly addressed in the discussion.

The authors present their results in a short and to the point fashion and support their claims with sufficient evidence. Some minor changes in Figures 3 and 6 could help the reader to better understand and interpret the presented data.

6. In Figure 3 the labels describing the time point should include a unit of time. For example, T(+90) should be T(+ 90 min) or it should be clarified in the Figure description (279 - 283).

7. In Figure 6 A the dashed line and solid line in the legend should be labelled to make sure the reader can get all the information needed to interpret the data at a glance.

8. In Figure 6 B the old data set is on the right and the new data set is on the left side, while in 6 D it is the opposite. I would suggest to always keeping the old data on the left and the new data on the right.

9. The description of the samples sizes (331 – 332 and 347 – 348) is quite difficult to understand and briefly explaining which numbers describe the old and new data set would improve the reader’s understanding immensely.

10. In the new data set, the time to escape (Figure 6 C) seems to be more similar between the different mazes than in the old data set, which showed a clear difference between the maze types. This should be addressed and discussed briefly.

Briefly addressing points 5 and 10 as part of a very short discussion of the limitations of this study, could complete the otherwise thorough and compelling discussion.

Reviewer #2: This fun, accessible paper was a treat to read. I do have a few thoughts as to how to improve the manuscript: the authors need to decide if they are writing a “how to do it” paper or a paper seriously engaged in the biology of growth patterns: the different growth patterns and strategies of fungi and why fungi may adopt different strategies. I’m guessing the authors intend a protocols/”how to do it” paper because a great deal of literature about fungal networks and network biology is completely absent (e.g., heaps of papers by Mark Fricker). IF the authors really want to engage in explanations for discovered patterns, please know there is a lot written about fungal networks and the mathematics and evolution of different network strategies. Not to acknowledge that work here, if the intent is to seriously engage in a discussion of growth patterns, is painful.

As a methods paper, the authors claim that a standardized system of measuring fungal growth would be advantageous for the discipline, as it is difficult to compare growth of fungi across experiments without a common set of protocols and agreed standards. Hyphal velocity is reported widely in the literature, but branching patterns and fine-scale growth strategies (e.g. through mazes) are less commonly reported.

That’s a compelling argument, but again the authors are lacking citations of much of the primary literature describing research on fungal growth rate metrics. To measure fungal hyphal growth rate and branching without acknowledging the historical (or even recent) work that has been done in the field is to miss some of the relevance of the newly described work. Many sentences in the final paragraph of the introduction allude to this work without directly citing it.

Previous researchers who have done similar experiments on hyphal growth rate and branching include (but are not limited to) Morrison and Righelato 1974, Prosser and Tough 1991, and Camenzind et al 2020. A relationship between hyphal growth rate and branching rate has already been reported and seems a counter to the results reported here (lines 337-338).

Referring to data on the growth velocity of A. nidulans and other isolates (which isolates?) with multiple runs , the authors suggest that there may be natural variation in growth velocity for a given isolate (lines 427-432). However, nutrient accessibility can enhance or reduce the hyphal growth rate of fungi (Camenzind et. al 2020; Prosser and Tough 1991; Morrison and Righelato 1974). Media recipes were not consistent across labs, correct? Can the authors dismiss nutrient accessibility/supply as a factor that could cause variation in hyphal growth velocity? This may be less of a concern than we think because the Momany lab ran all A. nidulans replicates, but the A. nidulans mutants did require adjustments to the media recipe. Either way, it would be good to know which isolates the authors reference and more about what’s going on here.

In general, for a methods paper, the methods need to be better explained.

One of the aims of this experiment appears to be to standardize the methods researchers use to quantify fungal growth (lines: 79-80, 94-169). However, the preparation of isolates in this experiment is highly variable across labs. Researchers grew different strains of the same fungal species at different temperatures and used different media recipes. They used different dilutions for the spore slurries and injected the microfluidic devices with different amounts of inoculum. Some isolates incubated in the microfluidic device before imaging, while the imaging for other isolates started immediately. Did the collaborators try and keep the concentration of inoculated spores consistent across the isolates? Why were the media recipes variable between isolates of the same species, especially when there is an extensive literature on what conditions maximize growth for most of the species in this experiment? When did the researchers determine when to start imaging? Is the variation in the fungal preparation methods due to the authors framing the experiment as a competition? These discrepancies caused confusion for this reader who expected a paper about standardized methods and they need to be addressed.

For example, for the Fungus Preparation section, please include the same information in each description, in other words, if you tell the reader the relevant lab for one fungus, include that information for all fungi. Strive to make these descriptions consistent. Are all relevant citations included? E.g. no citation is given for Medium N, is it such a standard medium that all readers will know exactly how to make it? This is a paper that is supposed to standardize protocols. Please give all information for all experiments so that a reader would be able to replicate them exactly.

Similarly, if someone wanted to print their own microfluidic devices, are templates or instructions available, if they are, how shall a reader find them?

The authors state: “Fungal species were selected by each team and loaded into devices using a general protocol shared with all participants.” Should that protocol be provided as supplementary material? Including the instructions each lab received for the fungal olympics would potentially answer many of the above questions.

In the results section, the authors explain that nubs are a unique feature fo R.stolonifer’s growth (Lines 333-337, 354-355). They mention that nubs increased with decreasing hyphal velocity. While the authors discuss nubs extensively in the data analysis section, they don’t explain the role of nubs in fungal development. What is the authors’ justification for studying nubs? What conclusions can the authors draw about the role of nubs in R.stolonifer’s development?

The authors found no relationship between the formation of branches and nubs and the time to escape for R. stolonifer (Lines 355-356, figure 7 part C). In figure 7 part C, it looks like the formation of nubs and the time to escape the maze are positively correlated. How did the authors come to the conclusion that branches and nub formation do not correlate to the time to escape the maze? More detail is needed here, perhaps indication of R^2 values.

Other comments: What’s meant by the word “lifestyle”? Do the authors mean life history strategy, growth pattern, ecological niche, something else? I see this word quite a bit in papers now but it seems to mean lots of different things and it would be useful to define it. As used it’s too vague to be meaningful.

Line 80: “towards increasing” is awkward. Writing is slightly awkward in places, throughout.

In the abstract, the authors include the straight channel as a microfluidic design (4 microfluidic designs total), but in the body of the paper (line 252) they explain that the straight-line feature is part of every microfluidic device. This minor discrepancy should be addressed.

Key citations (but please also look at papers that have cited these papers!):

Morrison, K.B., R.C. Righelato.1974. "The Relationship Between Hyphal Branching, Specific Growth Rate and Colony Radial Growth Rate in Penicillium chrysogenum".81: 517-20 DOI: 10.1099/00221287-81-2-51

Prosser, J.I, A.J. Tough. 1991. "Growth Mechanisms and Growth Kinetics of Filamentous Microorgansims". Critical Reviews in Biotechnology. https://doi.org/10.3109/07388559109038211

Camenzind, Tessa, Anika Lehmann, Janet Ahland. Stephanie Rumpel, and Matthias C. Rillig. 2020. “Trait-based approached reveal fungal adaptations to nutrient-limiting conditions.” Environmental Microbiology. 22 (8): 3548-3560 doi: 10.1111/1462-2920.15132. Epub 2020 Jul 8. PMID: 32558213.

6. PLOS authors have the option to publish the peer review history of their article (what does this mean?). If published, this will include your full peer review and any attached files.

Reviewer #1: No

Reviewer #2: No

---

## [Author Response · Author response to Decision Letter 0]

15 Jun 2021

• We would like to thank the reviewers for reading our work and for their constructive critiques. We have carefully revised the manuscript to reflect reviewer’s feedback and our point by point responses are highlighted below.

• We have revised the manuscript to reflect journal formatting requirements.

• We clarified in the manuscript that the accession numbers for the clinical isolates could not be provided due to the restrictions on sharing data. This statement is included in the methods section and data availability statements.

• We corrected the competing interest section, which now mentions that Paul Guerette, Edyta Szewczyk, Kevin McCluskey and David Breslauer are employees of Bolt Threads Inc.

• We have added the following to our funding statement: “Bolt Threads Inc provided support in the form of salaries for authors PG, ES, KM and DB but did not have any additional role in the study design, data collection and analysis, decision to publish, or preparation of the manuscript. The specific roles of these authors are articulated in the ‘author contributions’ section.”

• We have added the updated statements to the cover letter. 

• The map in figure 1 was drawn by author FE using Adobe Illustrator and based on online maps in the public domain (USGS National Map Viewer) which provided general proportions.

• We thank the reviewer for pointing this out. The wide variety of fungi used has complicated the selection of a term which accurately encompasses all the participating organisms. We have decided that cfu/mL is the most accurate and converted all units to this throughout the methods section (Lines 91-245) of manuscript. 

• We have added this information to the methods section for each respective species. Lines 139-140, 158-

• We have added this information to the Neurospora crassa section of the methods (Line 145).

There was no set time limit used for exclusion. Fungi simply needed to reach the halfway point by the time the timelapse taken for the experiment had ended. We have added this information to that section for clarification (Lines 220-225).

• As this event was framed as a competition, each group was encouraged to select their own “optimal” conditions for their fungal entry. We have added text to clarify this as well as to mention the impact nutrient availability from differing culture conditions may have on the results (lines 204-207, 380-381, 437-439).

• We have added this information to the figure legend for Figure 3.

• The lack of labeling for the solid and dashed line legend was an error and we thank the reviewer for pointing it out. We have added back the labels to Figure 6A.

• We have adjusted Figure 6B to match the suggestions here.

• We have adjusted the wording and added indicators of “new” and “old” to help clarify the description here (lines 343-345).

• We thank the reviewer for pointing this out, however it is difficult to clarify the source of the heterogeneity without a substantially larger number of experiments. We have, however, added some discussion of possible sources of heterogeneity in results to the text to reflect the limitations of this 

• We thank the reviewer for pointing out this gap—as written, this omission would indeed have been painful. We have clarified the purpose of the study in the introduction (lines 79-92) as follows: Despite the clear advantages of microfluidics for the measurement of hyphal growth rates and branching patterns many fungal research groups do not have access have the engineering expertise needed to design and manufacture the devices or to the microscopy equipment needed to accurately record the results. Where microfluidic devices have been used, current methodologies are often highly customized for each species and vary significantly between labs. In order to make microfluidics with live imaging easily accessible to research groups working on a range of fungi, we started a worldwide crowd-sourced collaborative effort, the Fungus Olympics. 

• And in the Discussion lines 419-424 as follows:

• Though we did not set out to examine hyphal branching and exploration strategies, something that has been the subject of study by other groups ( reviewed by Harris, 2008, Heaton et al 2012, and Prosser and Tough 1991), the differences in exploration strategies of T. versicolor and R. stolonifer were striking.

• The reviewer raises a good point about the possibility that nutrient accessibility might be important (not just in the medium, but also as a result of metabolism). To address this point, we added a third possible explanation for the differences in branching pattern in the Discussion (lines 437-439):

• It is also possible that differences in nutrient availability and metabolism might have contributed to different growth patterns as has recently been shown at the colony level for several grassland saprotrophic fungi (Carmenzid et al, 2020). 

• We clarified that standardization refers to the measurement of phenotypes. Standardizing the fungi culture conditions was not part of the goals. A general protocol for the use and loading of each device (now provided as a supplement) was given to all participants as a unified starting point. Some steps needed to be optimized for each fungi, as different fungal growth rates would determine how long it took each fungi to grow out of the central loading chamber and to reach the maze features and researchers needed to start imaging before their fungi entered the device features. Variation in media and incubation temperature was due to the competitive aspect of this work, where each group chose what they expected would be optimal for their selected fungal species/strains. We have added further text to highlight this (lines 204-207, 380-381).

• We have added the relevant labs for each fungus to the methods for consistency. We have also added further method details and media citations to allow for reproducibility for each experiment. For Medium N specifically, the citation was in reference #16. We now mention this reference immediately after the mention of Medium N to avoid any confusion. General methods for the production of the Fungus Olympics microfluidic devices are available in the device fabrication section. Those interested in this specific design could contact the Irimia lab through the email address provided as corresponding author. 

• We have added the general set of instructions that was provided to each group as a supplement to this work (S1 Fig).

• We don’t have enough data to say what the nubs are doing. We have added the following statement on line 351: though the role of nubs is not clear, it seems possible that they might be branch initials that fail to extend when hyphal velocity slows.

• While there does appear to be a positive correlation between the formation of nubs and time to escape for some of the mazes, this difference was not significant. Future experiments designed to specifically probe this question may reveal more details into this relationship. We have added text to the results and discussion to clarify this (Lines 351-353, lines 429-432).

• The final sentence has been shortened to remove “lifestyle”.

• Future work will more systematically examine these trends.

• We have fixed this line. 

• All 3 different maze designs and the straight channels are present in each microfluidic device. We have added text to the legend in Figure 1 and altered the text in the abstract to clarify that this is a single device with 4 different features, not 4 individually designed devices. Lines 261-262.

• We thank the reviewer for pointing out this omission. We have added reference to these papers in the discussion along with one from Fricker (Heaton et al 2012) and one from Harris (Harris, 2008). References #38, 39.

---

## [Decision Letter · Decision Letter 1]

9 Jul 2021

PONE-D-21-05622R1

Crowdsourced Analysis of Fungal Growth and Branching on Microfluidic Platforms

PLOS ONE

Dear Dr. Irimia,

Thank you for submitting your manuscript to PLOS ONE. After careful consideration, we feel that it has merit but does not fully meet PLOS ONE’s publication criteria as it currently stands. Therefore, we invite you to submit a revised version of the manuscript that addresses the points raised during the review process. Specifically, reviewer 2 had a small number of comments that should not take long to address and I will accept the article as soon as the changes have been made. 

We look forward to receiving your revised manuscript.

Kind regards,

Richard A Wilson

Academic Editor

PLOS ONE

Journal Requirements:

Reviewers' comments:

Reviewer's Responses to Questions

**Comments to the Author**

1. If the authors have adequately addressed your comments raised in a previous round of review and you feel that this manuscript is now acceptable for publication, you may indicate that here to bypass the “Comments to the Author” section, enter your conflict of interest statement in the “Confidential to Editor” section, and submit your "Accept" recommendation.

Reviewer #1: All comments have been addressed

Reviewer #2: (No Response)

2. Is the manuscript technically sound, and do the data support the conclusions?

Reviewer #1: Yes

Reviewer #2: Yes

3. Has the statistical analysis been performed appropriately and rigorously? 

Reviewer #1: Yes

Reviewer #2: Yes

4. Have the authors made all data underlying the findings in their manuscript fully available?

Reviewer #1: Yes

Reviewer #2: Yes

5. Is the manuscript presented in an intelligible fashion and written in standard English?

Reviewer #1: Yes

Reviewer #2: Yes

6. Review Comments to the Author

Reviewer #1: (No Response)

Reviewer #2: Response to revisions:

Thank you to the authors for their thorough response to our suggestions. By emphasizing the competitive aspect of the Fungal Olympics, the authors make it clear they are standardizing the fungal growth metrics and maze design only. The methods section is more organized and the author’s description of the microfluidic device makes it easier to visualize the experimental setup.

The authors made some effective changes to the discussion section. Framing the differences in growth strategies as unexpected strengthens their argument that microfluidic devices are useful for fungal biology. The authors also provide some interesting explanations for the morphology of R.stolinofer. I liked that the authors mention that they can use microfluidic devices to further explore these questions. Additionally, thank you for addressing that differences in nutrient availability and metabolism may have contributed to the differences in growth strategies. This is a necessary part of the discussion and fits nicely into the paper.

Minor issues and suggestions:

Lines 46-47: The authors made the suggested revision to omit the word ‘lifestyle’ here, but the abstract on page 1 of the manuscript is not revised. The authors should update the abstract on page 1 as well.

Lines 79-82: “Despite the clear advantages of microfluidics for the measurement of hyphal growth rates and branching patterns, many fungal research groups do not have access to have the engineering expertise needed to design and manufacture the devices or to the microscopy equipment needed to accurately record the results.”

There is a small error in this sentence that should be clarified.

Lines 98-99: In the methods section, the authors usually spell out the media recipe before abbreviating, except two instances. The authors should abbreviate the media recipe they used only after writing out the full name.

Line 148: Thank you to the authors for converting all units to cfu/mL. It greatly improves the clarity of the methods section. Line 148 is the only place in the methods that the concentration has not been converted to cfu/mL.

Supplementary Figure 1 (Loading protocol): The units for the inoculum are in cells/mL. The concentration should be in cfu/mL to be consistent with the unit in the methods section of the paper.

7. PLOS authors have the option to publish the peer review history of their article (what does this mean?). If published, this will include your full peer review and any attached files.

Reviewer #1: No

Reviewer #2: No

---

## [Author Response · Author response to Decision Letter 1]

1 Sep 2021

We would like to thank the reviewers for reading our work, their positive responses to our revisions and for their continued constructive critiques. We have carefully revised the manuscript to reflect reviewer’s feedback and our point by point responses are highlighted below.

Minor issues and suggestions:

Lines 46-47: The authors made the suggested revision to omit the word ‘lifestyle’ here, but the abstract on page 1 of the manuscript is not revised. The authors should update the abstract on page 1 as well.

We apologize for the error and have updated the abstract in the submission system to match our updated manuscript abstract.

Lines 79-82: “Despite the clear advantages of microfluidics for the measurement of hyphal growth rates and branching patterns, many fungal research groups do not have access to have the engineering expertise needed to design and manufacture the devices or to the microscopy equipment needed to accurately record the results.”

There is a small error in this sentence that should be clarified.

We have updated this sentence. “The clear advantages of microfluidics for the measurement of hyphal growth rates and branching patterns are attracting many fungal research groups to use these technologies. However, where microfluidic devices have been used, current methodologies are often highly customized for each species and vary significantly between labs.”

Lines 98-99: In the methods section, the authors usually spell out the media recipe before abbreviating, except two instances. The authors should abbreviate the media recipe they used only after writing out the full name.

We have updated these lines to contain the full names of the media used, to match the rest of the methods section.

Line 148: Thank you to the authors for converting all units to cfu/mL. It greatly improves the clarity of the methods section. Line 148 is the only place in the methods that the concentration has not been converted to cfu/mL.

We have updated this line to correct the units to cfu/mL to match the unified units in the rest of the document.

Supplementary Figure 1 (Loading protocol): The units for the inoculum are in cells/mL. The concentration should be in cfu/mL to be consistent with the unit in the methods section of the paper.

We have updated the figure to now be cfu/mL, in line with the unified units in the manuscript.

---

## [Editor Report · Decision Letter 2]

13 Sep 2021

Crowdsourced Analysis of Fungal Growth and Branching on Microfluidic Platforms

PONE-D-21-05622R2

Dear Dr. Irimia,

We’re pleased to inform you that your manuscript has been judged scientifically suitable for publication and will be formally accepted for publication once it meets all outstanding technical requirements.

Kind regards,

Richard A Wilson

Academic Editor

PLOS ONE
---

## [Editor Report · Acceptance letter]

20 Sep 2021

PONE-D-21-05622R2 

Crowdsourced Analysis of Fungal Growth and Branching on Microfluidic Platforms 

Dear Dr. Irimia:

I'm pleased to inform you that your manuscript has been deemed suitable for publication in PLOS ONE. Congratulations! Your manuscript is now with our production department. 

Kind regards, 

on behalf of

Dr. Richard A Wilson 

Academic Editor

PLOS ONE